

# Potential ecosystem service delivery by endemic plants in New Zealand vineyards: successes and prospects

Morgan W. Shields[1], Jean-Marie Tompkins[2], David J. Saville[3], Colin D. Meurk[4] and Stephen Wratten[1]

[1] Bio-Protection Research Centre, Lincoln University, Lincoln, Canterbury, New Zealand
[2] Environment Canterbury, Lincoln, Canterbury, New Zealand
[3] Saville Statistical Consulting Limited, Lincoln, Canterbury, New Zealand
[4] Landcare Research, Lincoln, Canterbury, New Zealand

Corresponding author
Morgan W. Shields,
morgan.shields@lincolnuni.ac.nz

## ABSTRACT

Vineyards worldwide occupy over 7 million hectares and are typically virtual mono-cultures, with high and costly inputs of water and agro-chemicals. Understanding and enhancing ecosystem services can reduce inputs and their costs and help satisfy market demands for evidence of more sustainable practices. In this New Zealand work, low-growing, endemic plant species were evaluated for their potential benefits as Service Providing Units (SPUs) or Ecosystem Service Providers (ESPs). The services provided were weed suppression, conservation of beneficial invertebrates, soil moisture retention and microbial activity. The potential Ecosystem Dis-services (EDS) from the selected plant species by hosting the larvae of a key vine moth pest, the light-brown apple moth (*Epiphyas postvittana*), was also quantified. Questionnaires were used to evaluate winegrowers' perceptions of the value of and problems associated with such endemic plant species in their vineyards. Growth and survival rates of the 14 plant species, in eight families, were evaluated, with *Leptinella dioica* (Asteraceae) and *Acaena inermis* 'purpurea' (Rosaceae) having the highest growth rates in terms of area covered and the highest survival rate after 12 months. All 14 plant species suppressed weeds, with *Leptinella squalida*, *Geranium sessiliforum* (Geraniaceae), *Hebe chathamica* (Plantaginaceae), *Scleranthus uniflorus* (Caryophyllaceae) and *L. dioica*, each reducing weed cover by >95%. Plant species also differed in the diversity of arthropods that they supported, with the Shannon Wiener diversity index ($H'$) for these taxa ranging from 0 to 1.3. *G. sessiliforum* and *Muehlenbeckia axillaris* (Polygonaceae) had the highest invertebrate diversity. Density of spiders was correlated with arthropod diversity and *G. sessiliflorum* and *H. chathamica* had the highest densities of these arthropods. Several plant species associated with higher soil moisture content than in control plots. The best performing species in this context were *A. inermis* 'purpurea' and *Lobelia angulata* (Lobeliaceae). Soil beneath all plant species had a higher microbial activity than in control plots, with *L. dioica* being highest in this respect. Survival proportion to the adult stage of the moth pest, *E. postvittana*, on all plant species was poor (<0.3). When judged by a ranking combining multiple criteria, the most promising plant species were (in decreasing order) *G. sessiliflorum*, *A. inermis* 'purpurea', *H. chathamica*, *M. axillaris*, *L. dioica*, *L. angulata*, *L. squalida* and *S. uniflorus*. Winegrowers surveyed said that they probably would deploy endemic plants around their vines. This research demonstrates that enhancing plant diversity in vineyards can deliver SPUs, harbour

ESPs and therefore deliver ES. The data also shows that growers are willing to follow these protocols, with appropriate advice founded on sound research.

## INTRODUCTION

Biodiversity and ecosystem-function relationships are a key component of agroecology, and agriculturalists are assisted by understanding how to deploy and manage functional diversity in the most appropriate ways. A key question in agroecology is the extent to which ecosystem services (ES) can be quantified and enhanced (*MEA*, *2005*; *Mooney*, *2010*; *Allan et al.*, *2015*; *Sandhu et al.*, *2015*; *Sandhu et al.*, *2016*). ES are defined as goods and services such as biological control that provide the foundation for sustaninaning human life on Earth (*Wratten et al.*, *2013*). The pathway for ES delivery includes the Service Providing Unit (SPU), defined as a the smallest unit, population or community that provides ES or will provide it in the future, within a given area (*Luck, Daily & Ehrlich*, *2003*). An Ecosystem Service Provider (ESP) is defined as the species, foodweb, habitat or system that faciliates and supports the provision of ES by an SPU (*Kremen*, *2005*). For example, a strip of flowering buckwheat, *Fagopyrum esculentum* Moench. and the predators and parasites which it supports can deliver multiple ES, including enhanced biological control of insect pests (*Scarratt, Wratten & Shishehbor*, *2008*).

Enhancing ES, SPUs and ESPs may be achieved by a better understanding of how biodiversity and its functions can contribute to reduced variable costs, sustainable agricultural production, agro-ecotourism and human wellbeing, among other factors (*Wratten et al.*, *2013*). Biodiversity delivers ecosystem functions (*Mooney & Ehrlich*, *1997*; *Swift & Anderson*, *2012*) and many of these functions have value for humans, thus becoming ES (*Cardinale et al.*, *2012*; *Mace, Norris & Fitter*, *2012*). The value of ES is increasingly being quantified to justify the incorporation of biodiversity into farming practices (*Fiedler, Landis & Wratten*, *2008*; *Tscharntke et al.*, *2012*; *Tuck et al.*, *2014*; *Barral et al.*, *2015*). *In situ* plant conservation continues to have a key role (*Keesing & Wratten*, *1997*) but with accelerating global biodiversity loss, policies and practices which enhance biodiversity in agricultural landscapes are increasingly important (*Wratten et al.*, *2013*). In that context, the provision of benefits by non-crop, low-growing, endemic New Zealand plants is quantified here and prospects for end-user adoption are assessed.

Worldwide, vineyards occupy over 7 million hectares (*The Wine Institute*, *2012*). Typically they are virtual monocultures of *Vitis vinifera* L. with bare earth or mown ryegrass (*Lolium perenne* L.) between the rows and sometimes with a few other spontaneous weed species (*Nicholls, Altieri & Ponti*, *2008*). Ryegrass and forb plants are also sometimes deliberately sown below vines, as in some organic vineyards (*Reeve et al.*, *2005*). It is well established that deployment of non-native biodiversity in vine inter-rows can enhance at least one ES, that of pest biocontrol (*Berndt, Wratten & Hassan*, *2002*; *Scarratt, Wratten & Shishehbor*, *2008*) but vegetation endemic to the country involved may provide a wider

range of ecosystem derived benefits, including reduced soil erosion from increased ground cover and soil moisture (*Ramos, Benito & Martínez-Casasnovas, 2015*), compared with the usual practice of herbicide-treated under-vine areas, conservation and eco-tourism, as well as cultural values (*Fiedler, Landis & Wratten, 2008*). Here, experimental field work investigated the potential of 13 endemic and one non-endemic, native plant species to provide ES in vineyards. For the purposes of this study, all the selected plant species are termed 'endemic.'

To evaluate the usefulness and benefits to growers of this approach, winegrowers were sent a questionnaire to elicit their perceptions of the barriers they face to deploy low-growing plants in vineyards. These data provided the study not only with future research directions but also practical insights on how best to achieve grower uptake. This socio-ecological aspect is a crucial step so that the pathway for agroecology research is comprehensive and is more likely to be accepted (*Warner, 2007*).

## MATERIALS AND METHODS

### Field experiment: environment and layout

The trial was located in the Waipara region, North Canterbury, New Zealand (E2489521: N5782109, altitude: 76 m) within the rows of grapevines (cv. Pinot Noir; 2.3 m inter-row width). Mean annual rainfall at the site was 684 mm, mean January (summer) temperature was 23 °C and soil type was Glasnevin silty loam (*Jackson & Schuster, 2002*). The field work, begun in October 2007, was a randomised complete block design comprising ten blocks, each with one replicate of 15 treatments. Each treatment comprised of 14 selected plant species and a control. The latter was maintained as bare earth by hand weeding. Such a control was used because in conventional viticulture worldwide, normal weed management practice comprises prophylactic use of herbicides under vines. The work carried out here was conducted in a conventional vineyard, therefore the control treatment employed regular weed removal. Each block consisted of four rows, each with 12 individual vines. Each experimental plot had two individual plants of one species of each species (or no plants in the case of the control): one on either side of a vine, about 30 cm from the trunk, arranged along the irrigation drip line. Replicates were separated by two vines in each row and vines were 1.5 m apart. Within-row management consisted of hand weeding in all plots every 2 weeks or when required prior to the weed suppression assessment. Inter-row management consisted of mowing the perennial ryegrass (*L. perenne*) every two weeks. The whole experiment utilised an area of the vineyard which was allocated by the company. No further space was available so plot size had to be restricted to the area around a single vine. Although this has implications for invertebrates moving between treatments, the latter were separated by two vines within a row and by an inter-row distance of 2.3 m, the latter comprising dense turf of perennial ryegrass. Table 1 lists plant species used in the trial and indicates the ES which were delivered or had potential for delivery.

### New Zealand plant species tested

Plant species were selected based on their growth habit (1–15 cm in height) to minimise interference with vine management. Species were further selected based on their shallow

**Table 1** Endemic[a] plant species used in the vineyard trial and the ecosystem associated benefits assessed.

| Plant species | Family | Ecosystem associated benefits | | | | |
|---|---|---|---|---|---|---|
| | | ES | ES | ES | ESP | EDS |
| | | Weed suppression | Invertebrate conservation[c] | Improving soil quality | Enhancing predator densities[c] | Pest development |
| *Acaena inermis* | Rosaceae | + | + | | + | |
| *Acaena inermis* 'purpurea'[b] | Rosaceae | + | + | + | + | + |
| *Anaphalioides bellidioides* | Asteraceae | + | + | | + | + |
| *Disphyma australe* | Mesembryan-themaceae | | + | | + | |
| *Geranium sessiliflorum* | Geraniaceae | + | + | + | + | + |
| *Hebe chathamica* | Plantaginaceae | + | + | + | + | + |
| *Leptinella dioica* | Asteraceae | + | + | + | + | + |
| *Leptinella squalida* | Asteraceae | + | + | | + | |
| *Lobelia angulata* | Lobeliaceae | + | + | + | + | + |
| *Muehlenbeckia ephedroides* | Polygonaceae | | + | | + | |
| *Muehlenbeckia axillaris* | Polygonaceae | + | + | + | + | + |
| *Raoulia hookeri* | Asteraceae | + | + | | + | + |
| *Raoulia subsericea* | Asteraceae | | + | | + | |
| *Scleranthus uniflorus* | Caryophyll-aceae | + | + | | + | + |

**Notes.**

[a] All plant species in this work apart from *M. axillaris* are endemic to New Zealand.

[b] A natural variation of *A. inermis* which has purplish coloration.

[c] Three sampling dates occurred, with some plant species sampled only once (*D. australe*, *M. ephedroides* and *R. subsericea*).

roots, floral characteristics and tolerance to frost, exposure, sun, drought and disturbance as well as practicalities such as cost and availability. All selected plants apart from *Muehlenbeckia axillaris* (Hook.f.) Endl. (also native to Australia) were New Zealand endemic species and all were perennial. Successful growth and survival of the plants were seen as prerequisites for their ability to provide sustainable benefits to the vineyard operation. Consequently, these parameters were assessed 6, 12 and 24 months after planting.

## Weed suppression

In September 2008, 11 months after planting, hand weeding was stopped in five selected blocks where the weed suppression assay was occurring. Normal vineyard management prevented cessation of weeding in the other five blocks, so they were excluded from this part of the overall experimental analysis. In December 2008, 14 months after planting, weed suppression by the plants was assessed visually by placing a 20 cm × 20 cm quadrat over them and over the corresponding area in the control plots (where none of the selected plants were planted) in the five selected blocks. Percentage cover of the study plants and weeds was recorded. *Disphyma australe* (subsp. Australe) Aiton, *Muehlenbeckia ephedroides* Hook.f. and *Raoulia subsericea* Hook.f. were not assessed due to their poor condition, growth and survival. Data were statistically analysed using a randomised block analysis of variance (ANOVA), followed by the unprotected Least Significant Difference (LSD) procedure at $P = 0.05$ (*Saville*, *1990*).

## Invertebrate biodiversity conservation

In August 2008 and January and March 2009 (10, 15 and 17 months, respectively, after establishment of the plants) under-vine treatments were assessed for invertebrate diversity and abundance using a suction sampler (*Arnold*, *1994*). In August 2008, samples were taken from the 14 plant treatments, the control and from the mid-point of inter-row areas (predominantly *L. perenne*) adjacent to the experimental plots in each of the ten blocks. The sampler was set on maximum power for 10 s, within which time an area of 0.04 m$^2$ was sampled at each location. Collected invertebrates were stored in 70% ethanol before being brought to the laboratory for sorting and identification. Due to gaps in formal taxonomic definitions, individuals were assigned to RTUs (recognisable taxonomic unit) for statistical analysis of diversity and abundance. For the second and third sampling dates, *R. subsericea*, *M. ephedroides* and *D. australe* were not sampled nor analysed because of their poor growth and survival. The Shannon Wiener diversity index ($H'$) was used because it takes into account evenness and species richness (*Magurran*, *1988*). Spiders are key predators of vineyard pests (*Thomson & Hoffmann*, *2007*), therefore spider density was analysed separately. Data were statistically analysed using a randomised block ANOVA, followed by the unprotected LSD procedure at $P = 0.05$.

## Soil quality

The effect of plant species on soil moisture and microbial activity was assessed. Due to resource constraints, only six plant species (those with the greatest growth and survival) were assessed. These were *Geranium sessiliforum* Simpson et Thomson, *Hebe chathamica* Cockayne et allan, *Leptinella dioica* Hook.f., *M. axillaris* and *Lobelia angulata* G. Forst. Control plots (bare earth) were also assessed.

Soil microbial activity was assessed by the TCC method (see *Alef & Nannipieri*, *1995*). This measures the rate of reduction of triphenyltetrazolium chloride (TTC) to triphenyl formazan (TPF) (*Alef & Nannipieri*, *1995*). It is a non-specific enzyme assay which determines the dehydrogenase activity in the soil and thereby indicates one aspect of soil microbial activity. In December 2008, soil samples were taken from below the five plant species listed above and the control plots in the five randomly selected blocks used in 'Weed suppression.' Within each plot, three 50 g subsamples of soil were collected at a depth of approximately 12 cm from around the roots of the selected plants, or within the corresponding area in the control plot, were combined to make a 150 g soil sample per plot. These 150 g soil samples were kept at 4 °C, before being assessed for microbial activity on the following day using the TTC method. The soil sampling method used above, was repeated in December 2008 and September and November 2009 for determination of soil moisture percentage. In the above six plant and control treatments, this was calculated using a gravimetric method and expressed on a dry weight basis (*Topp, Parkin & Ferré*, *1993*). Data for both soil parameters were statistically analysed using a randomised block ANOVA, followed by the unprotected LSD procedure.

## Pest development and longevity on candidate plants

The larval development of *E. postvittana* on the vegetative parts of the plant species was recorded in a laboratory bioassay. Species supporting high larval development rates could

potentially exacerbate pest problems in the vineyard by acting as a suitable host. However, there is also the possibility that these species could act as trap plants (*Khan et al.*, *2008*). Ten treatments including nine of the selected under-vine plant species and presentation of an artificial diet (*Shorey & Hale*, *1965*) were tested. Some plant species were not included in this bioassay as they had poor growth and/or survival in the field trial and were unlikely to be considered suitable for vineyard deployment; they were *M. ephedroides*, *R. subsericea* and *D. australe*. Others were excluded because another species or sub-species of the same genus was included in the bioassay; these were *Leptinella squalida* Hook.f. and *Acaena inermis* Hook.f. Six newly emerged (<24 h) first-instar larvae were placed in each of six Petri dishes (15 × 120 mm) in each of ten treatments. Treatments comprised freshly cut plant material with shoots inserted into an Eppendorf tube filled with water. Each tube was placed in a Petri dish which was sealed with plastic food wrap to prevent larval escape. After seven days, plant material was examined and water changed or the plant replaced as necessary. The artificial diet treatment consisted of cut squares of the diet substrate on which first instar-larvae were placed. There were six replicates of each treatment (a total of 6 × 6 = 36 larvae per treatment), arranged in a randomised block design under a 16:8 L/D photoperiod at 20 °C ±3. The number of larvae surviving to each development stage (second instar, third instar, final instar, pupa and adult) was recorded. A generalised linear model with a binomial distribution was used to determine the effect of treatment and development stage on *E. postvittana* survival.

## A questionnaire to winegrowers

Experimental work on ecosystem services enhancement in agriculture is of limited practical value unless agriculturalists are provided with ESPs (*Kremen*, *2005*) or similar to facilitate growers' adopting the work. To assess the likelihood of the latter, a questionnaire was mailed to 56 Waipara vineyard operators. Growers were asked "Which of the following uses of endemic plants would you consider adopting?" (see 'Winegrower questionnaires'). Growers were also asked "To what extent do the following factors lead you NOT to use endemic plants in or around your vineyard in the above ways?" (see 'Winegrower questionnaires'). This information was used to ensure that recommendations to growers were feasible and to identify future research directions.

## RESULTS

### Growth and survival of the selected plants

Significant differences in coverage (compared to the initially planted area) between plant treatments were found after 6 and 12 months (Table 2). *L. dioica* and *A. inermis* 'purpurea' showed greatest growth after 12 months while *Anaphalioides bellidioides* Glenny, *M. ephedroides*, *R. subsericea* and *D. australe* had little or no growth. After 24 months, survival remained high (≥90% ) for *M. axillaris*, *L. dioica*, *Raoulia hookeri* Allan var. hookeri, *A. inermis* 'purpurea' and *G. sessiliflorum* while that of other plants had begun to decline

### Weed suppression

There was significantly more weed growth in the control compared to all plant treatments ($P < 0.05$) (Fig. 1). *L. squalida*, *G. sessiliforum*, *H. chathamica*, *Scleranthus uniflorus* P.A.

**Table 2** Mean change in cover (m²) of endemic plant species from planting to 6 or 12 months, respectively, and their survival beneath grapevines at 12 and 24 months, respectively (for full species names see Table 1).

| Endemic plant[a] | Change in cover (m²) after: | | Survival (%) at: | |
|---|---|---|---|---|
| | 6 months | 12 months[b] | 12 months | 24 months |
| L. dioica | 0.24 | 0.38 | 100 | 100 |
| A. inermis 'purpurea' | 0.28 | 0.34 | 100 | 90 |
| L. angulata | 0.30 | 0.22 | 100 | 70 |
| L. squalida | 0.10 | 0.20 | 95 | 50 |
| G. sessiliflorum | 0.10 | 0.16 | 100 | 90 |
| M. axillaris | 0.20 | 0.15 | 100 | 100 |
| H. chathamica | 0.19 | 0.14 | 100 | 80 |
| R. hookeri | 0.13 | 0.13 | 100 | 100 |
| S. uniflorus | 0.06 | 0.13 | 100 | 80 |
| A. inermis | 0.07 | 0.12 | 60 | 60 |
| A. bellidioides | 0.06 | 0.04 | 90 | 40 |
| M. ephedrioides | 0.03 | 0.00 | 80 | 0 |
| R. subsericea | −0.03 | −0.03 | 60 | 10 |
| D. australe | 0.44 | −0.14 | 0 | 0 |
| **LSD(5%)[c]** | **0.10** | **0.12** | – | – |

**Notes.**
[a] All plant species in this work apart from *M. axillaris* are endemic to New Zealand.
[b] The table has been sorted into the order of decreasing growth to 12 months.
[c] LSD, Least Significant Difference. Means which differ by more than the LSD(5%) are significantly different at $P < 0.05$.

Will and *L. dioica* induced the greatest weed suppression (Fig. 1). Weeds consisted primarily of *Trifolium* spp. (Fabaceae) but also included Poaceae, Malvaceae and Asteraceae families.

## Invertebrate biodiversity conservation

At all sampling dates there was a significant effect of treatment on invertebrate diversity and there was greater overall abundance in the summer (January and March) than in winter (August) (Table 3). A total of 3,133 invertebrate individuals from 16 taxa were collected over all the sampling dates. During summer (January and March 2009), Hemiptera (1,936 individuals), Araneae (203) and Formicidae (175) were the most abundant taxa. In winter (August 2008), Araneae (72), Diplopoda (54) and Diptera (37) the dominant.

During early summer (January 2009), *M. axillaris*, *G. sessiliflorum*, *A. bellidioides*, *L. dioica*, *L. squalida*, *L. angulata*, *A. inermis* and *R. hookeri* had significantly higher diversity than either of the controls ($P < 0.05$) (Table 3). In late summer (March 2009), *M. axillaris*, *G. sessiliflorum*, *A. inermis* 'purpurea' and *L. angulata* had significantly greater diversity than the ryegrass inter-row control ($P < 0.05$), while these and *A. inermis*, *H. chathamica*, *A. bellidioides*, *L. squalida* and *L. dioica* had significantly higher diversity indices than the bare earth control ($P < 0.05$) (Table 3). In winter (August 2008), *G. sessiliflorum*, *H. chathamica*, *A. bellidioides*, *A. inermis* 'purpurea,' *L. dioica*, *M. axillaris* and *L. squalida* had significantly higher invertebrate diversity than either of the controls (bare earth and ryegrass inter-row treatments) ($P < 0.05$) (Table 3).

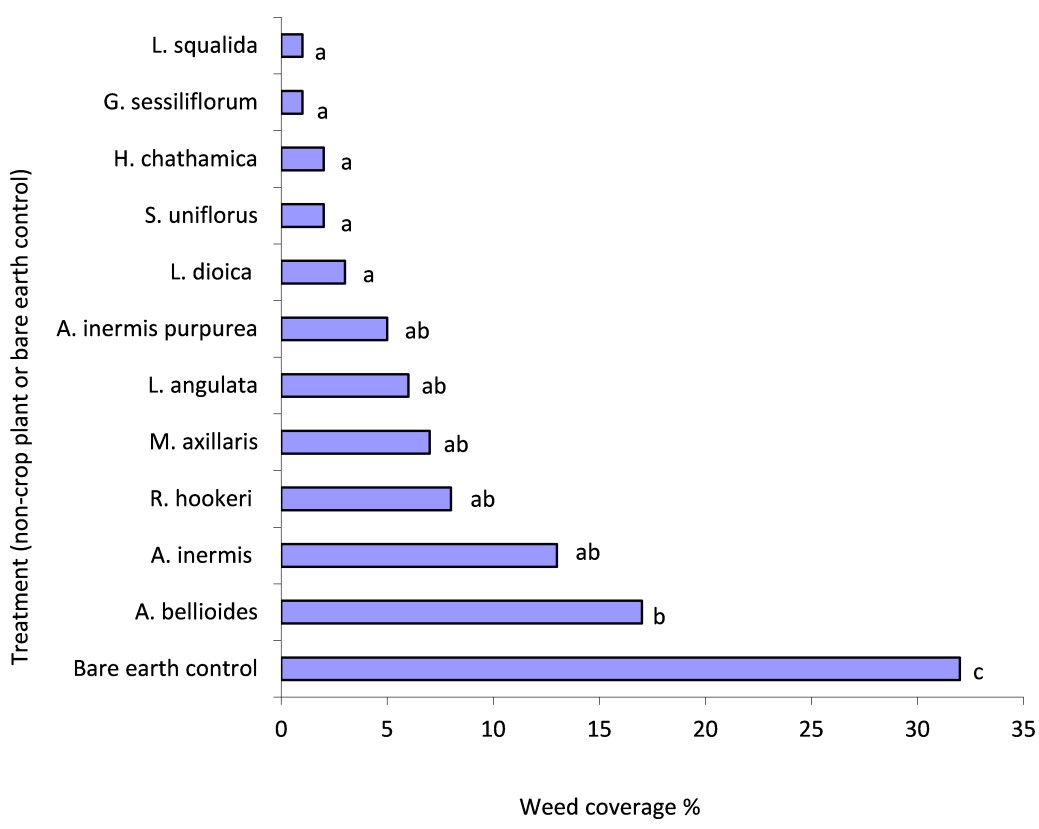

**Figure 1** **Mean weed penetration of under-vine treatments within the 0.04 m² areas assessed.** Treatments with a letter in common are not significantly different from one another at $P < 0.05$. Letters were assigned using the unprotected LSD procedure (*Saville*, *1990*); LSD(5%) = 13.

A significant effect of treatment on spider density was found for all sampling dates, with highest spider abundance in March 2009 (Table 4). Spider density was significantly correlated with arthropod diversity on the August and March sampling dates.

*G. sessiliflorum* and *H. chathamica* consistently had the highest densities of spiders. *A. inermis* 'purpurea,' *A. bellidioides*, *L. angulata* and *M. axillaris* also had significantly higher spider densities than did the bare earth control treatment on at least one of the sampling dates.

Spider families included web-building spiders in the Theridiidae (Sundervall), Linyphiidae (Blackwall), Agelenidae (Koch) and Amaurobiidae (Thorell) families. Wandering/hunting spider families included Oxyopidae (Thorell), Salticidae (Blackwall), Gnaphosidae (Pocock), Clubionidae (Wagner) and Pisauridae (Simon).

## Soil quality—moisture and microbial activity
### Soil moisture
Soil moisture in the bare earth control treatment was low relative to the other treatments on all three sampling dates (Table 5). In September and November 2009, it was also low under the *L. dioica* treatment. In November 2009, it was significantly higher below *L. angulata* and *A. inermis* 'purpurea' compared to all other treatments ($P < 0.05$) (Table 5).

**Table 3  Mean Shannon–Wiener diversity indices for invertebrates in under-vine treatments at three sampling dates, ranked for 2008 results.** Treatments with means of 0 have been omitted from the analysis of variance, as denoted by placing these means in brackets. The variability of such treatments is zero, so a LS Effect (5%) has been calculated to allow comparison between bracketed and unbracketed means (for full species names see Table 1).

| Endemic plant[a] | Invertebrate diversity (Shannon-Weiner $H'$) | | |
|---|---|---|---|
| | Aug 2008[b] | Jan 2009 | Mar 2009 |
| G. sessiliflorum | 1.11 | 1.17 | 1.31 |
| H. chathamica | 0.95 | 0.24 | 0.77 |
| A. bellidioides | 0.71 | 1.10 | 0.57 |
| A. inermis 'purpurea' | 0.45 | 0.55 | 1.10 |
| L. dioica | 0.35 | 1.09 | 0.50 |
| M. axillaris | 0.28 | 1.30 | 1.31 |
| L. squalida | 0.26 | 0.98 | 0.52 |
| L. angulata | 0.17 | 0.94 | 1.01 |
| A. inermis | 0.15 | 0.92 | 0.79 |
| D. australe | 0.07 | – | – |
| M. ephedrioides | 0.07 | – | – |
| R. hookeri | 0.07 | 0.71 | 0.24 |
| R. subsericea | 0.07 | – | – |
| S. uniflorus | (0) | (0) | 0.07 |
| Ryegrass inter-row | (0) | 0.19 | 0.43 |
| Bare earth | (0) | 0.07 | (0) |
| **LSD(5%)[c]** | **0.36** | **0.49** | **0.45** |
| **LSEffect(5%)[d]** | **0.25** | **0.34** | **0.32** |

**Notes.**
[a] All plant species in this work apart from *M. axillaris* are endemic to New Zealand.
[b] The table has been sorted into the order of decreasing Shannon–Wiener $H'$ mean values in August 2008.
[c] LSD, Least Significant Difference. Unbracketed means which differ by more than the LSD (5%) are significantly different at $P < 0.05$.
[d] LSeffect, Least Significant Effect. If a bracketed mean and an unbracketed mean differ by more than the LS Effect(5%), then the two means are significantly different at $P < 0.05$.
–, means plant species was not sampled.

### Soil microbial activity

Microbial activity in December 2008 was higher in all the plant treatments compared to the bare earth control, while it was significantly higher beneath *L. dioica* compared to that under the other plant treatments ($P < 0.05$) (Table 5). Although soil moisture may influence microbial activity, it was very low in all treatments at the time of microbial activity assessment.

## Development of *E. postvittana* larvae on the selected plant species

There was a significant effect of plant species ($P < 0.001$) and the larval instar reached ($P < 0.001$) on survival of the pest *E. postvittana*, but there was no significant interaction between treatment and instar ($P = 0.99$) (Fig. 2). Survival across all stages was significantly higher on the artificial diet than on any of the plant species used, suggesting that the selected plants provided sub-optimal nutrition to *E. postvittana*. *E. postvittana* larval survival was significantly higher on *A. inermis* 'purpurea' than on any of the other tested plants. The

**Table 4** **Mean density of spiders/m$^2$ for different under-vine endemic plant treatments in August 2008, January 2009 and March 2009.** Treatments with means of 0 or 3 (one spider in one plot) have been omitted from the analysis of variance, as denoted by placing these means in brackets. The variability of such treatments is nil or very low, so assuming it is zero, an LS Effect (5%) has been calculated to allow comparison between bracketted and unbracketted means (for full species names see Table 1).

| Endemic plant[a,b] | Density of spiders/m$^2$ in: | | |
|---|---|---|---|
| | **Aug 2008** | **Jan 2009** | **Mar 2009** |
| *L. dioica* | 8 | (0) | 5 |
| *A. inermis* 'purpurea' | 15 | 10 | 45 |
| *L. angulata* | (0) | 33 | 20 |
| *L. squalida* | (3) | 10 | (0) |
| *G. sessiliflorum* | 60 | 38 | 83 |
| *M. axillaris* | 20 | 8 | 30 |
| *H. chathamica* | 38 | 45 | 70 |
| *R. hookeri* | (0) | 8 | 13 |
| *S. uniflorus* | (3) | (3) | (0) |
| *A. inermis* | 5 | 15 | 18 |
| *A. bellidioides* | 18 | 18 | 23 |
| *M. ephedrioides* | (0) | – | – |
| *R. subsericea* | (0) | – | – |
| *D. australe* | 10 | – | – |
| Ryegrass inter-row | (3) | 10 | 5 |
| Bare earth (control) | (0) | (3) | (0) |
| **LSD(5%)[c]** | **25** | **29** | **32** |
| **LSEffect(5%)[d]** | **18** | **20** | **23** |

**Notes.**
  [a] All plant species in this work apart from *M. axillaris* are endemic to New Zealand.
  [b] This table has been sorted into the same order of endemic plants as Table 2.
  [c] LSD, Least Significant Difference. Unbracketed means which differ by more than the LSD(5%) are significantly different at $P < 0.05$.
  [d] LSEffect, Least Significant Effect. If a bracketted mean and an unbracketted mean differ by more than the LSEffect(5%), then the two means are significantly different at $P < 0.05$.
  –, means plant species was not sampled.

other species supported decreasing pest survival in the order: *G. sessiliflorum*, *L. angulata*, *R. hookeri*, *L. dioica*, *M. axillaris*, *S. uniflorus*, *A. bellidioides* and *H. chathamica*. In the case of *H. chathamica*, no pest larvae survived to the adult stage.

## Overall ranking of endemic plant species

In Table 6, the 14 plant species are ranked for each of the characteristics summarised in Tables 2–5 and Figs. 1–2. For most characteristics, the plant species with the highest mean value is assigned the rank of 1. However, for weed suppression and leafroller (pest) survival, a rank of 1 is assigned to the species that had the fewest weeds or had the lowest pest survival.

Some plant species were not evaluated for all characteristics, often because they had already been judged unsuitable. Only six species were assessed in all respects (Table 6). None of these was consistently the best in delivering ES. For example, *L. dioica* ranked first for growth, survival and microbial activity, but ranked 10 out of 11 for spider density, and

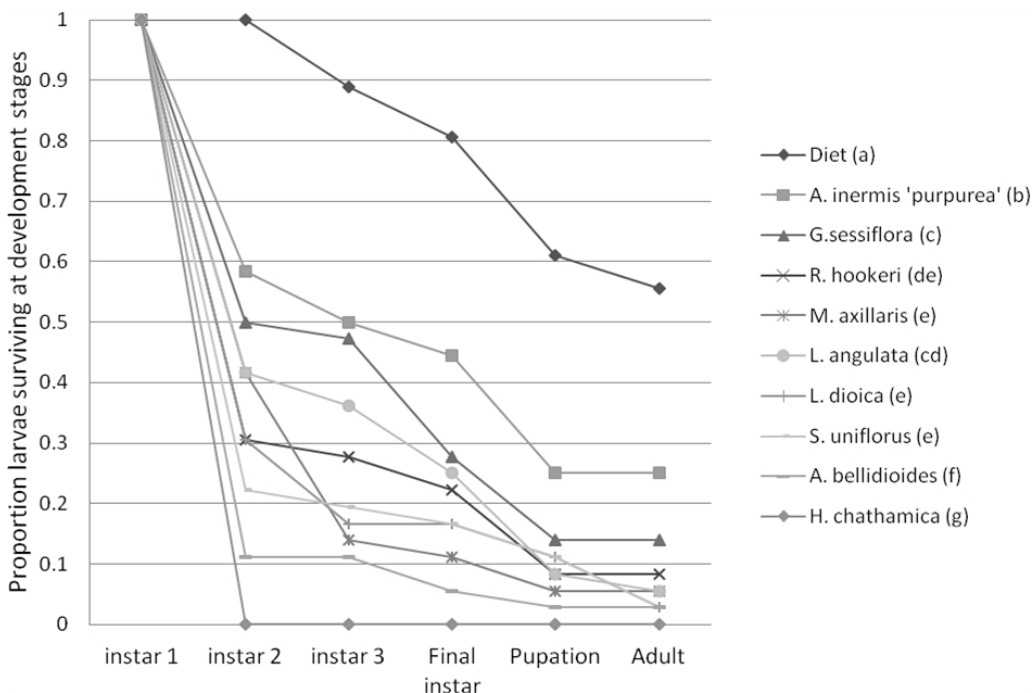

**Figure 2  Mean proportion of leafroller, *Epiphyas postvittana*, larvae surviving at each development stage.**  Treatment names which have a letter in common indicate the two treatments are not significantly different in overall survival (averaged over all development stages) at $P < 0.05$.

**Table 5  Mean soil moisture percentage for different under-vine treatments in December 2008, September 2009 and November 2009, and mean microbial activity as measured by the TTC method on the first date.** Soil moisture is expressed on a dry weight basis (for full species names see Table 1).

| Endemic plant[a,b] | Soil moisture (%) in: | | | Mean microbial activity (TTC method) [(rate of reduction of TTC, $\mu$g)/(g dry soil/hr)] |
|---|---|---|---|---|
| | Dec 2008 | Sep 2009 | Nov 2009 | |
| *L. dioica* | 6.5 | 11.6 | 8.3 | 20.0 |
| *A. inermis* 'purpurea' | 7.7 | 14.8 | 14.3 | 13.3 |
| *L. angulata* | 7.0 | – | 16.2 | 12.2 |
| *G. sessiliflorum* | 5.2 | 17.1 | 8.7 | 12.2 |
| *M. axillaris* | 6.4 | 17.6 | 8.9 | 11.6 |
| *H. chathamica* | 5.0 | 16.3 | 8.3 | 12.9 |
| Bare earth | 5.3 | 10.3 | 7.1 | 6.7 |
| **LSD(5%)[c]** | **2.6** | **4.0** | **5.0** | **4.4** |

**Notes.**

[a]All plant species in this work apart from *M. axillaris* are endemic to New Zealand.

[b]This table has been sorted into the same order of endemic plants as Table 2.

[c]LSD, Least Significant Difference. Means which differ by more than the LSD (5%) are significantly different at $P < 0.05$.

–, means plant species was not adequately sampled on this date.

Shields et al. (2016), *PeerJ*, DOI 10.7717/peerj.2042

**Table 6 Ranking of endemic plant species by change in growth, survival beneath grapevines and ecosystem associated benefits; weed suppression, mean invertebrate diversity, mean spider density, mean soil moisture, leafroller survival and microbial activity on one date.** A rank of 1 was the best in terms of desirability. A mean ranking was calculated for only those endemic plants for which all attributes had been assessed. Ties were replaced by mean ranks; e.g., three 1= values were replaced by 2s, and two 4= values by 4.5 s (for full species names see Table 1).

| Endemic plant[a] | Growth (m$^2$)[b] to 12 months | Survival (%) to 24 months | Ecosystem associated benefits | | | | | | Mean ranking |
|---|---|---|---|---|---|---|---|---|---|
| | | | ES | ES | ESP | ES | ES | EDS | |
| | | | Weed suppression at 11 months | Invertebrate diversity (Shannon–Wiener $H'$) | Density of spiders/m$^2$ | Soil moisture (%) | Mean microbial activity (TTC method) [(rate of reduction of TTC, μg)/(g dry soil)/hr] | Leaf-roller (pest) survival | |
| *L. dioica* | 1 | 1= | 5 | 7 | 10 | 6 | 1 | 4= | **4.6** |
| *A. inermis* 'purpurea' | 2 | 4= | 6 | 5 | 3 | 2 | 2 | 9 | **4.2** |
| *L. angulata* | 3 | 8 | 7 | 4 | 6 | 1 | 4 | 7 | **5.0** |
| *L. squalida* | 4 | 10 | 1= | 9 | 9 | – | – | – | – |
| *G. sessiliflorum* | 5 | 4= | 1= | 1 | 1 | 4 | 5 | 8 | **3.8** |
| *M. axillaris* | 6 | 1= | 8 | 2 | 4= | 3 | 6 | 4= | **4.5** |
| *H. chathamica* | 7 | 6= | 3= | 6 | 2 | 5 | 3 | 1 | **4.3** |
| *R. hookeri* | 8 | 1= | 9 | 10 | 8 | – | – | 6 | – |
| *S. uniflorus* | 9 | 6= | 3= | – | 11 | – | – | 3 | – |
| *A. inermis* | 10 | 9 | 10 | 8 | 7 | – | – | – | – |
| *A. bellidioides* | 11 | 11 | 11 | 3 | 4= | – | – | 2 | – |
| *M. ephedrioides* | 12 | 13= | – | – | – | – | – | – | – |
| *R. subsericea* | 13 | 12 | – | – | – | – | – | – | – |
| *D. australe* | 14 | 13= | – | – | – | – | – | – | – |

**Notes.**

[a]All plant species in this work apart from *M. axillaris* are endemic to New Zealand.

[b]The table has been sorted into the order of decreasing growth to 12 months.

–, means plant species was not assessed.

**Table 7** Current and potential use of endemic plants within Waipara vineyards (survey responses from $n = 30$ growers).

| Endemic plant ecosystem benefit use | Number of growers establishing endemic plant for ecosystem associated benefits listed on left[a] | | | | | | |
|---|---|---|---|---|---|---|---|
| | N/A | Already do this | Definitely | Maybe | Probably not | Definitely not | Already + Definitely |
| As groundcover to suppress weeds beneath vines | 0 | 2 | 3 | 20 | 4 | 1 | 5 |
| To provide resources to beneficial vineyard insects | 0 | 10 | 6 | 14 | 0 | 0 | 16 |
| To reduce soil erosion in the vineyard | 7 | 6 | 12 | 4 | 0 | 1 | 18 |
| To conserve beneficial invertebrates | 1 | 17 | 8 | 4 | 0 | 0 | 25 |
| To contribute to endemic plant conservation | 1 | 18 | 8 | 2 | 1 | 0 | 26 |
| For eco-marketing purposes | 9 | 7 | 6 | 6 | 2 | 0 | 13 |

**Notes.**
[a]Number of growers who currently or potentially would use endemic plants in the manner indicated.

7 out of 11 for invertebrate diversity. By comparison, *G. sessiliflorum* ranked first for weed suppression, invertebrate diversity and spider density, but ranked 8 out of 9 for leafroller (pest) survival.

When judged by an overall ranking, the most promising plant species were (in decreasing order) *G. sessiliflorum*, *A. inermis* 'purpurea,' *H. chathamica*, *M. axillaris*, *L. dioica* and *L. angulata*, with average ranks ranging from 3.8 to 5.0, respectively (Table 6). None of the other eight plant species averaged a rank of 5.0 or more, when their ranks were averaged over the characteristics for which they had been assessed.

## Winegrower questionnaires

The survey response rate was 30 out of 56 growers (Table 7). The majority of respondents (who had not already adopted endemic plants for any purpose) indicated that they would 'definitely' or 'maybe' deploy endemic plants around or within their vineyard properties for the various uses presented to them. Currently, the conservation of flora and fauna are the primary purposes of endemic plants within respondents' properties and they stated that such plants are also likely to be established for erosion control, enhancement of pest biological control or for weed suppression.

Growers were also asked to indicate whether certain factors had led them not to deploy endemic plants for the uses listed above (Table 8). These, which may be seen as barriers to establishing such plants for the various uses, included a lack of knowledge, cost of initial investment, risk, disruption to normal practices or having no interest in such practices (Table 8). For most endemic plant uses, the primary concern of growers was the initial investment required. Notably, however, a lack of knowledge surrounding the use of such plants to suppress weeds beneath vines was cited by an almost equal number of growers as was the barrier of initial investment. Risk was a barrier cited by nearly half the growers for establishing endemic vegetation for conservation of flora and fauna. Risk was also stated by a significant proportion of growers as cause for not utilising endemic plants for marketing purposes.

**Table 8  Potential barriers to deploying endemic plants within vineyard properties.** For each plant use, the number of respondents for which the use was applicable is given in the right-hand column.

| Endemic plant ecosystem benefit use | Number of growers citing barriers to establishing endemic plant for various uses | | | | | | |
|---|---|---|---|---|---|---|---|
| | N/A | Lack of knowledge | Initial investment | Risk | Disruption to normal practices | No interest by grower | Number of respondents to whom applicable |
| As groundcover to suppress weeds beneath vines | 0 | 12 | 11 | 4 | 4 | 2 | 30 |
| To provide resources to beneficial vineyard insects | 0 | 4 | 10 | 1 | 5 | 0 | 30 |
| To reduce soil erosion in the vineyard | 7 | 3 | 6 | 1 | 1 | 1 | 23 |
| To conserve beneficial invertebrates | 1 | 3 | 7 | 13 | 1 | 0 | 29 |
| To contribute to endemic plant conservation | 1 | 3 | 7 | 13 | 13 | 0 | 29 |
| For eco-marketing purposes | 9 | 3 | 5 | 14 | 14 | 1 | 21 |

## DISCUSSION

Findings here suggest the selected endemic plants deployed beneath vines have the potential to improve pathways to ES provision (i.e., SPU, ESP and ES themselves) ultimately improving value to growers. Overall, certain endemic plant species may preserve biodiversity, enhance biological control of vineyard pests, provide weed suppression and improve soil health. Clearly further research is required, such as repeating the trial in different regions. In the first trial described in this paper, however, the most promising plant species were *G. sessiliflorum*, *A. inermis* 'purpurea,' *H. chathamica*, *M. axillaris*, *L. dioica* and *L. angulata*.

### Weed suppression

Management of weeds is a major concern of vineyard managers as these plants can compete with the vines' surface 'feeder' roots for resources and can act as refuges for pests (*Tesic, Keller & Hutton*, *2007*; Waipara Valley North Canterbury Winegrowers, pers. comm., 2009). In this study, all the plant species assessed significantly suppressed weeds when compared to unplanted treatments. Whether suppression was sufficient to remove the need for further weed management would depend on the plant species deployed and the weed cover tolerances of individual growers. Plant cover and weed suppression were not significantly correlated, so while some plants may cover a large area, their growth form may not be dense enough to reduce weed penetration. The extent of weed pressure within the trial vineyard may be considered low (control plots had only 30% weed cover) compared to other vineyards with higher rainfall. Consequently, if endemic plant species are to be established in regions with higher weed pressure, suppression or management will need to be correspondingly more intensive to maintain a steady state with an appreciable presence of the endemic plants.

### Invertebrate biodiversity conservation

On all sampling dates, invertebrate diversity was higher for *G. sessiliflorum* than in the bare earth control or the ryegrass inter-row, whereas *M. axillaris* had the highest

invertebrate diversity in summer (Table 3). Overall, diversity was lower in winter, which is not surprising considering typical invertebrate phenology (*Dent & Walton*, *1997*; *Bowie et al.*, *2014*). However, invertebrate diversity levels were maintained over the winter period by *G. sessiliflorum*, *H. chathamica*, *A. bellidioides* and *A. inermis* 'purpurea' (Table 3), indicating that they provided suitable overwintering sites for invertebrates. This has implications for early-season pest biological control because early pest control by overwintering invertebrates may prevent pest outbreaks later in the season (*Ramsden et al.*, *2015*). While there is debate over the extent to which species richness correlates positively to ecosystem functioning (*Loreau, Naeem & Inchausti*, *2002*; *Cardinale et al.*, *2006*), it remains the case that the extent of ecosystem functions depends on the traits of the species examined and their sensitivity to environmental change, and diversity is most likely to provide greater functional potential and resilience.

## Conservation biological control (CBC)

Increasing plant diversity by adding beneficial plants has become a fundamental part of integrated pest management (IPM) theory and practice (*Bugg & Waddington*, *1994*; *Landis, Wratten & Gurr*, *2000*; *Gurr, Wratten & Snyder*, *2012*; *Ratnadass et al.*, *2012*). Increased rates of biological control under these conditions have often been attributed to the more diverse system providing natural enemies with resource subsidies including alternative food and shelter (*Landis, Wratten & Gurr*, *2000*; *Altieri & Nicholls*, *2004*; *Gurr, Wratten & Altieri*, *2004*; *Zehnder et al.*, *2007*; *Helyer, Cattlin & Brown*, *2014*). Also, diverse assemblages of arthropod taxa associated with some of the selected plant treatments (Table 3) included potential alternative prey such as Collembola, Diptera, Hemiptera etc. For example, spider densities were higher for several plant treatments than the controls. This is consistent with other research (*Thomson & Hoffmann*, *2007*) and was probably due to the plants providing suitable (and permanent) shelter. Spiders can reduce insect pest populations (*Marc, Canard & Ysnel*, *1999*; *Midega et al.*, *2008*) and in vineyards have been implicated as key predators of pests (*Hogg & Daane*, *2010*) including *E. postvittana*, mealybugs (*Pseudococcus* spp.), scales (Hemiptera: Coccidae) and mites (Acari: Eriophyidae) (*Thomson & Hoffmann*, *2007*). The most abundant spider families represented in this study included web-building Linyphiidae and Theridiidae and the wandering/hunting Salticidae and Oxyopidae (*Paquin, Vink & Duperre*, *2010*). These all predate *E. postvittana* and feed on both larval and adult stages of this pest (*MacLellan*, *1973*; *Danthanarayana*, *1983*; *Hogg et al.*, *2014*).

## Soil improvements

For all plant species, the estimated soil moisture was always similar to or higher than the control on all three sample dates. It is well established that competition for water between the crop and added plant biodiversity can be a major factor in farmers' agronomic decision making (*Warner*, *2007*). However, there was no obvious competition for water between the added plants and the vines, which obtain most of their water from deep roots, rather than surface 'feeder' roots (*Jackson*, *2000*). Soil biological activity increased beneath grapevines with endemic plant understoreys which may correspond to enhanced nutrient cycling (*Mader et al.*, *2002*) compared to the control. The identity of those

organisms responsible for such increases could now be addressed by the use of molecular methods (*Hirsch, Mauchline & Clark*, *2010*). The influence of the plants on the above parameters may increase over time, especially after further leaf litter accumulation and root development, although the dry conditions of many vineyards in summer (occupying largely 'Mediterranean' climates (*Hannah et al.*, *2013*)) may limit soil microbial activity (*Labeda, Kang-Chien & Casida*, *1976*).

### Potential of the selected plants to host the pest *E. postvittana*: an ecosystem dis-service (EDS)

Results suggested that some of the plant species could be suitable hosts to the larvae of this pest. However, of the three plant species identified (*L. dioica*, *A. inermis* 'purpurea' and *M. axillaris)* to be the most promising for vineyard deployment by their growth and floral resource, *L. dioica* and *M. axillaris* supported the lowest mean larval survival and, along with the other plants tested (Fig. 2), pose little threat of enhancing *E. postvittana* populations.

### Winegrower attitudes

The majority of growers indicated they would consider incorporating endemic plants into their properties (Table 7). However, several potential barriers to such action were identified and these would need to be overcome to achieve widespread establishment of endemic plants. These barriers centred on lack of knowledge of the other potential effects of plant establishment and the initial investment required (Table 8). This is probably because at the time of the survey, this practice was still in the research phase with protocols yet to be made available to winegrowers. Perceived risk was a notable barrier to growers establishing endemic plants in their vineyards (Table 8). This response is probably due to concerns that such vegetation may exacerbate bird damage to grapes by providing resources (shelter, food etc.) which may support pest bird populations (Waipara Valley North Canterbury Winegrowers, pers. comm., 2009).

### Evaluating the benefits provided by non-crop plants in vineyards

It is critical that the establishment of endemic plants in vines is financially viable. Market-based incentives may exist for provision of enhanced ES, such as weed suppression, pest control and marketing. For instance, premium prices or higher demand for wine from 'clean green' vineyards that promote biodiversity-friendly business. However, other ES that such plants provide may be public goods and lack any direct financial incentive to the grower; conservation, cultural value and aesthetics are examples. This involves paying for ecosystem services (PES) which have value beyond the farm (*Wratten et al.*, *2013*). Compensation for ES that are public goods would probably entail government incentives such as subsidies or tax reductions (*Kroeger & Casey*, *2007*) and could be delivered via agri-environment schemes such as those in the USA, UK and Europe, although these have achieved mixed results (*Kleijn & Sutherland*, *2003*; *Kleijn et al.*, *2006*).

## CONCLUSIONS

Endemic New Zealand plants beneath grapevines can provide multiple potential ecosystem services, including weed suppression, biodiversity conservation, soil improvement and conservation biological control. In some cases in the current work, the plants constituted RTUs and harboured ESPs. For example, the added plant populations were SPUs for services such as biological control, they enhanced ESPs such as spiders and provided ES in the form of weed suppression and enhanced soil quality, expressed as higher moisture and microbial activity. Winegrowers are likely to establish endemic plants within vineyards if perceived and real barriers to such action are overcome. These include growers' lack of knowledge, initial investment, risk and disruption to normal practices. Also, farmers learn about and adopt new practices in a range of ways, and social learning (*Warner, 2007*) is one of these. Orthodox teaching/technology-transfer methods rarely work (*Cullen et al., 2008*). This New Zealand work is highly relevant to other regions as the traits of the plants in this study are likely to be similar to other plant species in vineyard ecosystems worldwide. Also, although *A. inermis* is endemic to New Zealand, it is now available commercially in the UK and USA and as seeds in New Zealand (http://www.nzseeds.co.nz/contact-us). The work presented here addresses a key current challenge, which is to maintain or enhance productivity of agro-ecosystems in a sustainable way and to reduce external costs by increasing the role that ES can play on farmland, while at the same time maintaining ecological integrity in the cultural landscape. This is critical to not only fulfilling international agreements on biodiversity protection, but also for the commercial benefit of an authentic 'clean green' brand. Meeting these challenges has been called 'sustainable intensification' (*Garnett et al., 2013*; *Pretty & Bharucha, 2014*) and the current work, although not concerning food, contributes to that. It illustrates how simple enhancements of agricultural biodiversity can help translate ecosystem science into action, thereby supporting the goals of the intergovernmental science-policy platform on Biodiversity and Ecosystem Services (www.ipbes.net).

## ACKNOWLEDGEMENTS

Thanks to Mud House New Zealand (now Accolade Wines) for the generous provision of a field site and to Jean-Luc Dufour at that site for mentoring.

### Funding

We received financial support from the New Zealand Ministry of Business, Innovation and Employment (LINX 0303), the Bio-Protection Research Centre, Lincoln University. The funders had no role in study design, data collection and analysis, decision to publish, or preparation of the manuscript.

### Grant Disclosures

The following grant information was disclosed by the authors:
New Zealand Ministry of Business, Innovation and Employment: LINX 0303.
Bio-Protection Research Centre, Lincoln University.

## Competing Interests

Prof. Stephen D. Wratten is an Academic Editor for PeerJ. David J. Saville is an employee of Saville Statistical Consulting Limited.

## Author Contributions

- Morgan W. Shields wrote the paper, prepared figures and/or tables, reviewed drafts of the paper.
- Jean-Marie Tompkins conceived and designed the experiments, performed the experiments, analyzed the data, wrote the paper, prepared figures and/or tables, reviewed drafts of the paper.
- David J. Saville conceived and designed the experiments, analyzed the data, prepared figures and/or tables, reviewed drafts of the paper.
- Colin D. Meurk contributed reagents/materials/analysis tools.
- Stephen Wratten conceived and designed the experiments, reviewed drafts of the paper.

## Data Availability

The raw data has been supplied as Data S1.

## Supplemental Information

Supplemental information for this article can be found online at http://dx.doi.org/10.7717/peerj.2042#supplemental-information.

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
