# Peer review of "Potential ecosystem service delivery by endemic plants in New Zealand vineyards: successes and prospects"

_PeerJ, doi:10.7717/peerj.2042_

## Round 0.1 · original submission · Minor Revisions

This is an interesting and well-written paper that covers multiple aspects of ES including plants, arthropods, soil, and even perception in the social/agricultural context.

Both reviewers were positive or generally positive about the paper. And having read it, I agree with their assessments and ask the authors to provide a response and rebuttal.

In particular, both reviewers noted some concern with the layout and, thus, independence of the treatments in terms of the arthropod results. I fully agree with this and request that the authors consider carefully how to deal with and/or respond to this concern. Reviewer #2 gives some helpful suggestions in this regard.

·

Basic reporting

The manuscript presents results on assessing the value and benefit of a variety of native plants to enhance management of a key pest in vineyards. The team examined 14 plant species that could be sown under and between rows of grapes, and assessed the degree to which the plants suppressed weeds, supported a diversity of arthropod taxa, and retained soil moisture and microbial activity. Combining the various criteria, they then ranked the plant species for the contributions they can make to Ecosystem Services, with the intent to deliver these results to growers for implementation. In addition, they surveyed growers for their willingness to incorporate the recommendations into their vineyard plans, and determined the key concerns that need to be overcome for greater adoption of recommendations.
The manuscript is written very well, with appropriate background to establish the relevance of the study.

Experimental design

The research question was well defined and is very relevant to the expanding field of understanding ecosystem services. The study is designed well, methods were detailed and appropriate, and allowed collection of data and robust analysis of the results. The appropriate use of statistics (and inclusion of a statistician in the study) enabled the analysis of the results to produce results that can be delivered to growers. Because the individual components measured have the potential for interactions, the analysis was critical for teasing apart the contributions of individual treatments.

Validity of the findings

The results were presented clearly and conclusions were supported by the results. The inclusion of survey results make the contribution extremely relevant and very likely that the recommendations will be implemented by growers. All too often, excellent research cannot find its way into adoption because of the missing step of finding key ways to gain acceptance and understand the pitfalls. This study should find its results put into practice.

Additional comments

Only two criticisms, both minor: 1) the spatial separation of treatment plots may not have been sufficient to ensure that insects did not move among treatments -- indeed, the movement is considered in other publications to be critical to the provision of ecosystem services; however, the design did not significantly influence the results or the interpretation of the results; and 2) it was not clear that soil microbial action at the soil surface affected plant health, given the depth of the plant roots -- if the benefits were for other purposes, I missed that connection. Otherwise, I believe this is a nicely conducted study that adds to the growing body of work supporting Ecosystem Services.

Reviewer 2 ·

Basic reporting

No Comments

Experimental design

No Comments

Validity of the findings

No Comments

Additional comments

The article is overall well-written with some exceptions; below I provide suggestions for minor revisions/improvements.

Introduction:
In general the introduction comes off as a bit disorganized – first some abbreviations are defined, then further along the concepts corresponding to those abbreviations are defined – and then there is a brief mention of BEF. The latter is given relatively short shrift (two lines), given that it is presented as one of the main theoretical frameworks underlying the experiments.

I would modify the first sentence, as it suggests that the authors are primarily concerned with the question of which ES’s can be quantified, which doesn’t characterize the paper well. A better choice for the thesis topic sentence might be something like “BEF is a central theme of both basic and applied ecology, and it arguably finds its most important application in agricultural ecology, where growers grapple with questions about on-farm diversity interacts with weed management and pest control….”

As mentioned above, ecosystem services are mentioned in the first line, but not defined until line 58 – at the very least, it seems there could be a better way of introducing the ecological concepts rather than first defining the abbreviated shorthand and then going on to talk about them. Perhaps do both at once?

More specific Points:
Line 70: pls define “low-growing plants” – and indicate that this refers only to non-crop plants (if that is the case) to avoid confusion.

Line 71 seems like an afterthought and detracts from the strong descriptive sentence preceding it – I would delete it/relocate it to further along (say, end of next paragraph, to line 89).

Some typos need correcting: Line 56, 61.

The brief (four lines, starting line 85) discussion of stakeholder attitudes about employing the various suggestions stemming from this experiment is underdeveloped and again seems like an afterthought at the end of this paragraph. Although it is clear that the intention of the surveys is only to get an overview of grower attitudes (rather than a more sophisticated qualitative analysis), this survey and ensuing results turns out to be an important piece of the paper, and as such deserves its own paragraph.

Methods:
Lines 98-102: these sentences, with an aside to organic practices, is somewhat confusing; it would suffice to state simply that this work was done in conventional vineyard, and you’re comparing different interventions (plant species additions) with a control (no plants).

Lines 110-112: A simple diagram with the spatial layout of the experimental block(s) would be very helpful. From what I can tell, the close proximity of the treatments could be an issue (as the authors state) because arthropod samples from different treatments may not be independent. In particular, one might be concerned that treatments that attract more arthropods might “spill over” into other plots nearby and skew the results. The authors might consider doing a post-hoc test of this – for instance, one could imagine developing a simple index of how likely different treatments might be to attract arthropods (and/or spiders), based on the underlying theory (for spiders, for example, this could be the natural enemies hypothesis – with the expectation that more plants would attract more prey for spiders and thus attract more spiders). Then one could use a 2X2 contingency table to test whether or not treatments that are in close proximity to other treatments that have a higher than average “index of attraction” for arthropods in fact have a higher than average density/diversity of arthropods/spiders.

Line 152: The organization is a bit strange here – I think spiders do not need their own sub-heading, but rather could easily be incorporated into the previous paragraph (esp. since collecting methodology is the same) by saying something like “Because spiders are an important measure of predation, an indicator of the biological control ES, they were analyzed separately….”

Line 162: I would add a TCC reference to the first sentence, and reiterate what you are testing (“To the effects of….?”). One Line 168, some indication of the depth of soil samples would be helpful.

Discussion:
Section 4.2: might be worthwhile including a brief mention of inducible defenses here – though mites are not mentioned explicitly, some of the work by R. Karban and his students over the years, esp. work done in San Joaquin valley vineyards, might be applicable here.
Section 4.5: would be helpful to highlight any patterns we might detect among plants that were better herbivore hosts – in particular, are there interesting phylogentic relationships among the plants species that might be a good indicator of how best to choose suitable plants going forward? Are the secondary chemical compounds in the “successful” plants similar?

---

## Round 0.2 · accepted · Accept

Following a generally positive peer review and a response/rebuttal from the authors, this paper is now acceptable for publication in PeerJ.

I encourage the authors to make the review history of this paper public, as it adds extra value to the final product.